

# A six-microRNA signature can better predict overall survival of patients with esophagus adenocarcinoma

Tian Lan[1,2], Yunyan Lu[3], Zunqiang Xiao[1], Haibin Xu[2], Junling He[2], Zujian Hu[2] and Weimin Mao[4,5]

[1] The Second Clinical Medical College, Zhejiang Chinese Medical University, Hangzhou, Zhejiang, People's Republic of China
[2] Department of Breast Surgery, Hangzhou Hospital of Traditional Chinese Medicine, Hangzhou, Zhejiang, People's Republic of China
[3] Department of Cardiology, Hangzhou Xiaoshan First People's Hospital, Hangzhou, Zhejiang, People's Republic of China
[4] Zhejiang Key Laboratory of Diagnosis and Treatment Technology on Thoracic Oncology (Lung and Esophagus), Zhejiang Cancer Hospital, Hangzhou, Zhejiang, People's Republic of China
[5] Department of Thoracic Surgery, Zhejiang Cancer Hospital, Hangzhou, Zhejiang, People's Republic of China

Corresponding author
Weimin Mao,
maowm1218@163.com

## ABSTRACT

**Background:** The microRNAs (miRNAs) have been validated as prognostic markers in many cancers. Here, we aimed at developing a miRNA-based signature for predicting the prognosis of esophagus adenocarcinoma (EAC).
**Methods:** The RNA-sequencing data set of EAC was downloaded from The Cancer Genome Atlas (TCGA). Eighty-four patients with EAC were classified into a training set and a test set randomly. Using univariate Cox regression analysis and the least absolute shrinkage and selection operator (LASSO), we identified prognostic factors and constructed a prognostic miRNA signature. The accuracy of the signature was evaluated by the receiver operating characteristic (ROC) curve.
**Result:** In general, in the training set, six miRNAs (hsa-mir-425, hsa-let-7b, hsa-mir-23a, hsa-mir-3074, hsa-mir-424 and hsa-mir-505) displayed good prognostic power as markers of overall survival for EAC patients. Relative to patients in the low-risk group, those assigned to the high-risk group according to their risk scores of the designed miRNA model displayed reduced overall survival. This 6-miRNA model was validated in test and entire set. The area under curve (AUC) for ROC at 3 years was 0.959, 0.840, and 0.868 in training, test, and entire set, respectively. Molecular functional analysis and pathway enrichment analysis indicated that the target messenger RNAs associated with 6-miRNA signature were closely related to several pathways involved in carcinogenesis, especially cell cycle.
**Conclusion:** In summary, a novel 6-miRNA expression-based prognostic signature derived from the EAC data of TCGA was constructed and validated for predicting the prognosis of EAC.

## INTRODUCTION

Globally, esophagus cancer was ranked seventh among the leading types of cancers and sixth among the leading causes of cancer mortality in 2018 according to the Global Cancer Observatory (*Fitzmaurice et al., 2018*). Although the diagnosis and treatment strategies have been developed, this cancer remains a major problem due to insufficient information on its etiology, and the overall five-year survival rate for patients with esophageal cancer is 15–25% worldwide (*Pennathur et al., 2013*). Generally, two types of malignancies are diagnosed: adenocarcinoma (10%) and squamous cell carcinoma (90% of cases). The prevalence of esophagus adenocarcinoma (EAC) has rapidly increased over the past few decades (*Thrift & Whiteman, 2012*). The prognosis of EAC is poor and its 5-year overall survival rate is 30% (*Hirst et al., 2011*). Due to the poor outcomes of EAC, it is important to reveal the mechanisms leading to the occurrence and development of EAC. More biomarkers that can effectively predict the genesis, progress, and prognosis of EAC need to be found urgently.

MicroRNAs (miRNAs) are small noncoding RNA transcripts that are made of estimated 22 nucleotides (*Lujambio & Lowe, 2012*). The predominant function of miRNAs is to regulate protein translation by binding to target messenger RNAs (mRNAs), and inhibit mRNA translation (*Krol, Loedige & Filipowicz, 2010*). They have recently been validated and applied in diagnosis and prognosis of a variety of tumors, including hepatocellular carcinoma (*Parizadeh et al., 2019*), prostate cancer (*Moya et al., 2019*), and breast cancer (*Yerukala Sathipati & Ho, 2018*).

Many studies focused on miRNAs in patients with Barrett's Esophagus (*Leidner et al., 2012*; *Li et al., 2018*; *Revilla-Nuin et al., 2013*), a precursor lesion of EAC. Yet, the miRNA expression landscape in EAC is not clearly understood. Over the past few years, some studies reported the significant role of miRNAs in the molecular diagnosis and prognosis of EAC. A 4-miRNA expression profile score can provide a validated approach for predicting pathological complete response rates to neoadjuvant treatment in EAC (*Skinner et al., 2014*). In addition, a 3-miRNA (miR-99b and miR-199a_3p and _5p) signature was correlated with patient survival and occurrence of lymph node metastasis (*Feber et al., 2011*). However, these findings were based on a small number of patients.

The Cancer Genome Atlas (TCGA), a landmark cancer genomics program, is a reservoir of large-scale miRNA-sequencing datasets spanning 33 cancer types. In the present investigation, we constructed a prognostic risk score system on the basis of miRNA datasets from TCGA to predict the prognosis of EAC. Furthermore, pathway enrichment and gene oncology annotation analyses were performed to understand the probable cellular functions of mRNAs associated with this signature.

## MATERIALS AND METHODS

### RNA-seq and clinicopathological data of EAC patients

From TCGA data portal (https://portal.gdc.cancer.gov/), RNA-seq data and associated clinical information were downloaded in January 2019. The annotation information was provided by GENCODE datasets (www.gencodegenes.org). Given that some miRNAs

and mRNAs display little or no expression in some tissues or do not vary sufficiently, only those with raw count value >20 in more than 80% of samples were retained for further analysis. Once normalization by edgeR was completed, this was followed by conversion of the expression patterns of miRNAs and mRNAs to log2 (normalized value +1) in preparation for the subsequent processing. Samples with a <1-month censor time are removed, because they cannot be representative samples for analyzing prognostic factors. A total of 84 EAC subjects with the corresponding clinical data including age, gender, height, weight, race, alcohol history, Barrett's disease history, tumor size, lymph node status, metastasis status, and TNM stage were collected in this study (Table 1). The EAC patients' dataset contained 96 samples (84 EAC and 12 normal tissues) and 272 miRNAs. Since the data came from the TCGA database, no further approval was required from the Ethics Committee.

## Construction and validation of the miRNA risk score

Eighty-four patients were stratified to two categories in a random manner: training set = 42, test set = 42. Training set was analyzed to build a miRNA model that was later confirmed in test and entire sets. In the training set, we screen out miRNAs with a significant $p$-value less than 0.1 by using univariate survival analysis based on Cox proportional hazards of each miRNA. The least absolute shrinkage and selection operator (LASSO) is a generalized linear regression algorithm capable of variable selection and regularization simultaneously (*Gao, Kwan & Shi, 2010*). We determined the lambda by using the cross-validation routine cv.glmnet with an n-fold equal to 10. Least absolute shrinkage and selection operator was performed to reduce above selected prognostic miRNAs further and to construct the risk score system.

For determination of survival risks, a prognostic model was created on the basis of miRNA data as follows:

$$\text{Risk score} = \sum_{i=1}^{n} \beta i * gene\ i$$

β stands for the coefficient of the miRNA, and gene refers to miRNA expression value.

Using the median score in training set as the cutoff, we stratified the subjects to low-risk and high-risk groups. The Kaplan–Meier (KM) and log-rank methods were applied to compare the survival rate between the groups by using the R "survival" package. The time-dependent receiver operating characteristic (ROC) curve was plotted by using the R "timeROC" package to evaluate specificity and sensitivity of the miRNA expression-based prognostic signature. Thereafter, this signature was validated in test set and entire set. ROC and KM curves were also carried out to validate accuracy and feasibility of the miRNA model. Then stratified analysis based on clinical parameters was performed in the entire set. All ROC and KM curves were plotted with R (version 3.5.2), and $p < 0.05$ represented statistical significance.

## Gene set enrichment analysis

Subjects were stratified to two groups (high and low) based on the risk score of the 6-miRNA signature. We used gene set enrichment analysis (GSEA, http://software.broadinstitute.org/gsea)

**Table 1 Clinical characteristics of EAC patients.**

| Variables | Case, *n* (%) |
|---|---|
| Sample number | 84 |
| Age | |
| <60 | 29 (34.52) |
| ≥60 | 55 (65.48) |
| Gender | |
| Male | 72 (85.71) |
| Female | 12 (14.29) |
| Height | |
| <175 | 41 (48.81) |
| ≥175 | 38 (45.24) |
| NA | 5 (5.95) |
| Weight | |
| <85 | 46 (54.76) |
| ≥85 | 37 (44.05) |
| NA | 1 (1.19) |
| Race | |
| Asian | 1 (1.19) |
| White | 66 (78.57) |
| NA | 17 (20.24) |
| Event | |
| Alive | 46 (54.76) |
| Dead | 38 (45.24) |
| Alcohol history | |
| No | 27 (32.14) |
| Yes | 56 (66.67) |
| NA | 1 (1.19) |
| Barrett's disease | |
| No | 52 (61.91) |
| Yes | 26 (30.95) |
| NA | 6 (7.14) |
| Tumor size | |
| 1 | 21 (25.00) |
| 2 | 14 (16.67) |
| 3 | 45 (53.57) |
| 4 | 1 (1.19) |
| NA | 3 (3.57) |
| Lymph node status | |
| 0 | 21 (25.00) |
| 1 | 47 (55.95) |
| 2 | 6 (7.15) |
| 3 | 5 (5.95) |
| NA | 5 (5.95) |

| Table 1 (continued). | |
|---|---|
| **Variables** | **Case, *n* (%)** |
| Metastasis | |
| 0 | 57 (67.86) |
| 1 | 11 (13.09) |
| NA | 16 (19.05) |
| Stage | |
| I | 12 (14.29) |
| II | 24 (28.57) |
| III | 33 (39.29) |
| IV | 11 (13.09) |
| NA | 4 (4.76) |

**Note:**
NA, not available.

(*Subramanian et al., 2005*) to figure out potential functional annotations in the two groups. The BioCarta dataset (c2.cp.biocarta.v6.2.symbols.gmt) served as the reference gene set. False discovery rate < 0.05, enrichment score > 0.5 were set as the significance threshold.

## Functional enrichment analysis

Using the miRNA target prediction tool starBase (http://starbase.sysu.edu.cn/index.php), the target genes of the 6-miRNA signature were predicted based on five datasets, including TargetScan, PITA, miRmap, microT, and miRanda. Metascape is a free online platform having a large-scale set of functional annotation tools to understand biological mechanisms behind a large pool of genes (http://metascape.org/gp/index.html#/main/step1). We used Metascape to analyze functional enrichment of Kyoto Encyclopedia of Genes and Genomes (KEGG) pathway and Gene Ontology (GO) based on the prognostic target genes of miRNAs and visualized by R "gglot2" package.

## RESULTS

### The predictive 6-miRNA signature for the training set

The overall design and workflow of this study were presented in Fig. 1. According to the results of the univariate Cox regression analyses, 64 miRNAs associated with survival data were selected for patients with EAC (Table S1). The lambda value was set by using the lambda.min, which is the value of lambda giving minimum mean cross-validated error, and then six miRNAs with nonzero coefficients were defined (Fig. S1). Based on the LASSO Cox regression models, a risk score was determined for each subject according to 6-miRNA status: Risk score = $(-0.6089 \times$ hsa-let-7b) + $(-0.1974 \times$ hsa-mir-23a) + $(0.3369 \times$ hsa-mir-3074) + $(0.0294 \times$ hsa-mir-424) + $(0.2421 \times$ hsa-mir-425) + $(0.2435 \times$ hsa-mir-505).

In the training set, the patients with EAC were divided into a high-risk group and a low-risk group. The risk scores of patients were ranked, and the dotplot was developed for the survival status of each patient. Compared with the mortality of patients in the high-risk

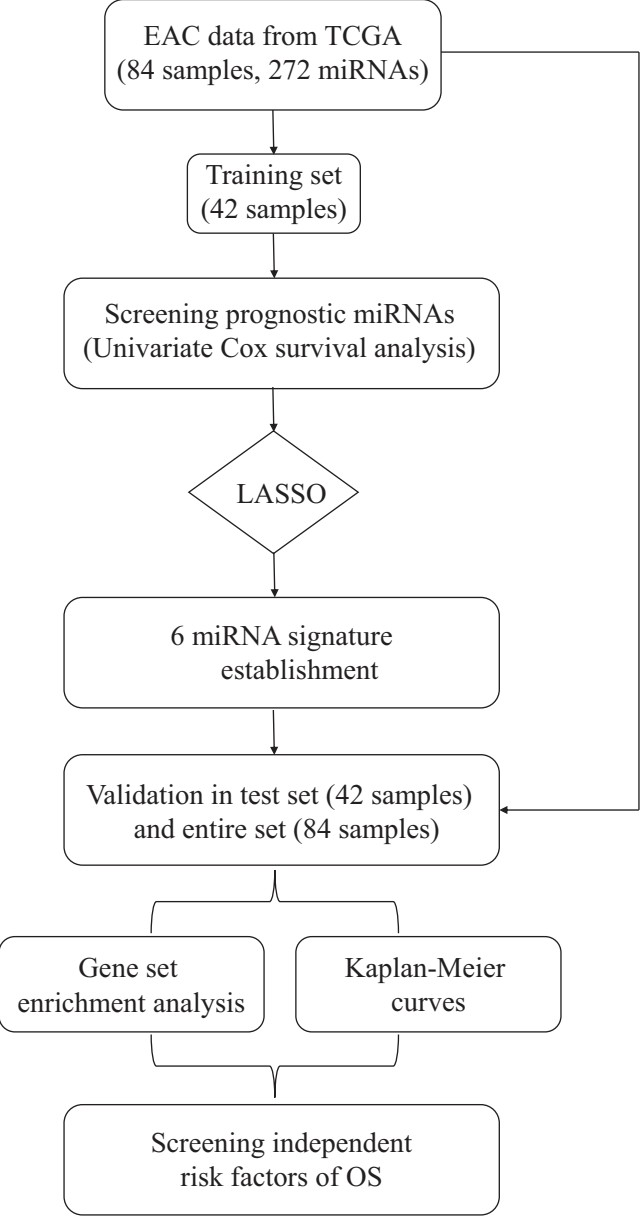

**Figure 1 Flow chart of data preparation, processing, analysis and validation in this study.**

group, those in the low-risk group was much lower (Figs. 2A and 2C). Moreover, based on a heatmap of the 6-miRNA profile, the levels of hsa-mir-3074, hsa-mir-424, hsa-mir-425, and hsa-mir-505 were lower in the low-risk group than those of the high-risk group. The level of hsa-let-7b and hsa-mir-23a were higher in the low-risk group than those of the high-risk group (Fig. 2D). The KM curve indicated that the survival time of patients in the high-risk group was shorter than those in the low-risk group (Fig. 2E). We described the predictive value of the 6-miRNA signature by using a time-dependent ROC curve. The area under curve (AUC) at 1, 2, and 3 years of the signature was 0.860, 0.962, 0.959, respectively (Fig. 2B).

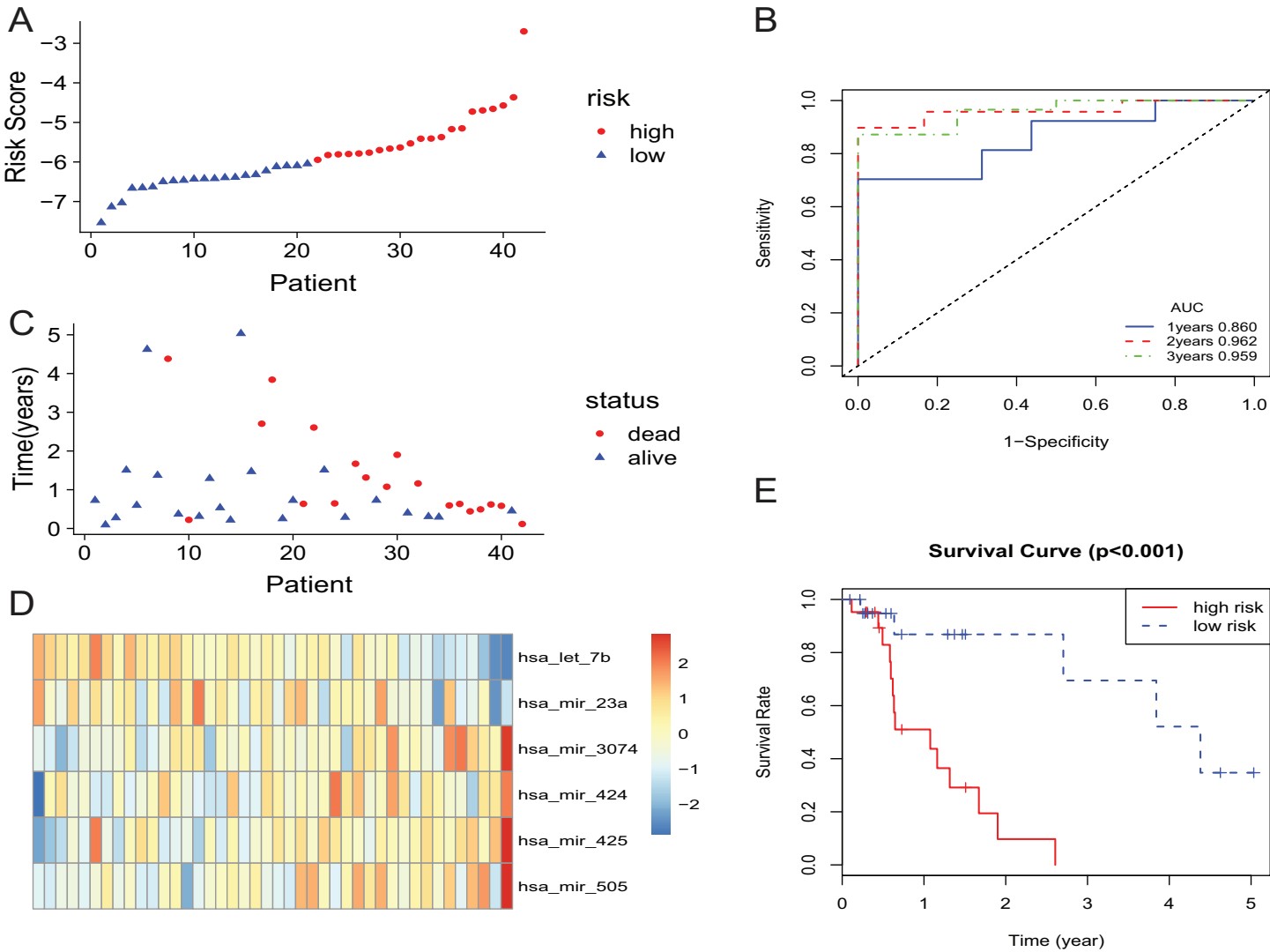

**Figure 2 The 6-miRNA signature predicted the OS of EAC patients in the training set.** (A, C) The 6-miRNA-based risk score and survival status of EAC patients. (B) Receiver-operating characteristic (ROC) analyzes the sensitivity and specificity of the survival time by risk score based on the 6-miRNA signature. (D) Expression heatmap of the 6 miRNAs corresponding to each sample which ranks in order of risk score. (E) Kaplan–Meier analysis for OS using the 6-miRNA signature.

## The predictive power of 6-miRNA signature in test set and entire set

The 6-miRNA signature was applied to the test set and the entire set for evaluation of its prognostic value. The distribution of risk scores, the expression values of six miRNAs and the survival status of patients ranked according to the risk scores were presented in test set (Figs. 3A, 3C and 3E) and entire set (Figs. 3B, 3D and 3F). In test set and entire set, patients with the low-risk scores exhibited better overall survival than those with the high-risk scores based on the KM curve (Figs. 3G and 3H). The 3-year AUC of the 6-miRNA-based signature was 0.840 and 0.868, respectively, for the test set and the entire set (Figs. 3I and 3J).

To assess the independent prognostic value of the 6-miRNA signature, various clinicopathological factors were subjected to univariate Cox regression and multivariate

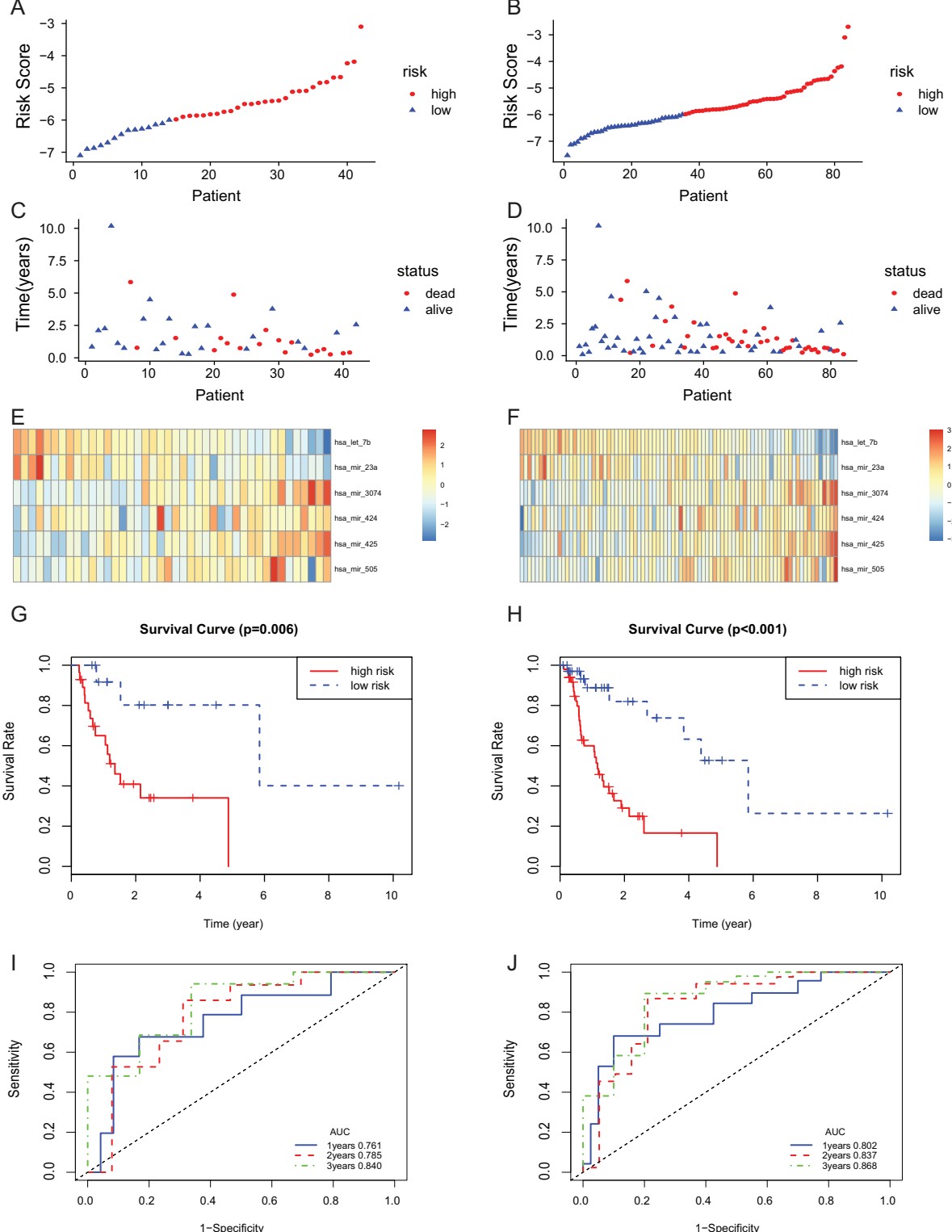

**Figure 3 The 6-miRNA signature predicted the OS of EAC patients in test and entire set.** The miRNA signature risk score distribution and heatmap of the miRNA expression profiles in test set (A, C, and E) and entire set (B, D, and F). survival curves of high- and low-risk samples in test set (G) and entire set (H). Time-dependent ROC curve for accuracy of the predicting risk score system in test set (I) and entire set (J).

**Table 2 Univariate and multivariate COX regression analyses of the six-microRNA signature and clinicopathologic factors in the entire set.**

| Variables | Univariate analysis | | | Multivariate analysis | | |
|---|---|---|---|---|---|---|
| | HR | 95% CI | *p*-value | HR | 95% CI | *p*-value |
| miRNA risk score | 3.41 | [1.70–6.84] | 0.001* | 2.95 | [1.43–6.07] | 0.003* |
| Age (≥60 vs <60) | 0.89 | [0.44–1.81] | 0.752 | | | |
| Gender (male vs female) | 0.68 | [0.20–2.31] | 0.539 | | | |
| Height (≥175 vs <175 cm) | 0.80 | [0.39–1.62] | 0.535 | | | |
| Weight (≥85 vs <85 kg) | 1.07 | [0.53–2.15] | 0.844 | | | |
| Alcohol consumption (yes vs no) | 0.46 | [0.23–0.92] | 0.029* | 0.67 | [0.32–1.40] | 0.287 |
| Barrett's disease (yes vs no) | 1.16 | [0.56–2.37] | 0.691 | | | |
| Stage (III+IV vs I+II) | 2.30 | [1.08–4.91] | 0.031* | 1.95 | [0.88–4.29] | 0.098 |

**Notes:**
HR, hazard ratio; 95% CI, 95% confidence interval.
* $p < 0.05$, statistically significant.

Cox regression analysis. The result indicated that the 6-miRNA signature was an independent prognostic factor after adjustment for other clinicopathological factors (HR = 2.95, CI [1.43–6.07], $p = 0.00338$, Table 2). When stratified by clinical factors (age, gender, caucasian, height, weight, alcohol consumption history, Barrett's disease, TNM stage), a nearly universal result was obtained for all subgroups (Fig. 4), showing that high-risk score was strongly associated with poor prognosis and vice versa. Regardless of height, weight, TNM stage, alcohol consumption history and Barrett's disease, the 6-miRNA signature is significantly effective. Therefore, the present results suggest that the 6-miRNA signature can predict the clinical prognosis of EAC.

## Functional analysis of the 6-miRNA signature

BioCarta pathway enrichment was conducted through GSEA in high-risk group in the entire set. It revealed that high-risk patients were associated with some pathways, including "proteasome pathway," "MCM pathway," "G2 pathway," and "cell cycle pathway" (Figs. 5A–5D). Through a miRNA prediction tool, starBase, 179 target mRNAs for hsa-let-7b, 147 for hsa-mir-23a, 382 for hsa-mir-424, 37 for hsa-mir-425, and 11 for hsa-mir-505 were obtained. Unfortunately, no target gene for hsa-mir-3074 was predicted. We conducted functional enrichment of these target genes by GO and KEGG categories. Cellular component, molecular function and biological process of these target genes based on $p$-values were showed (Figs. 5E–5G). The top 20 KEGG pathways of these target genes were plotted (Fig. 5H). Among these pathways, MAPK signaling pathway, hippo signaling pathway, foxo signaling pathway, and TGF-beta signaling pathway were reported to be related to metastasis of cancer (*Blum et al., 2019*; *Janse van Rensburg & Yang, 2016*; *Kim et al., 2018*; *Sun et al., 2018*). Some other pathways are also known to be associated with cancers, such as pathways in cancer, miRNAs in cancer, cell cycle, autophagy.

## DISCUSSION

Although great progress has been made in the field of the pathogenesis and clinical treatment of EAC, the overall morbidity and mortality for EAC have not improved

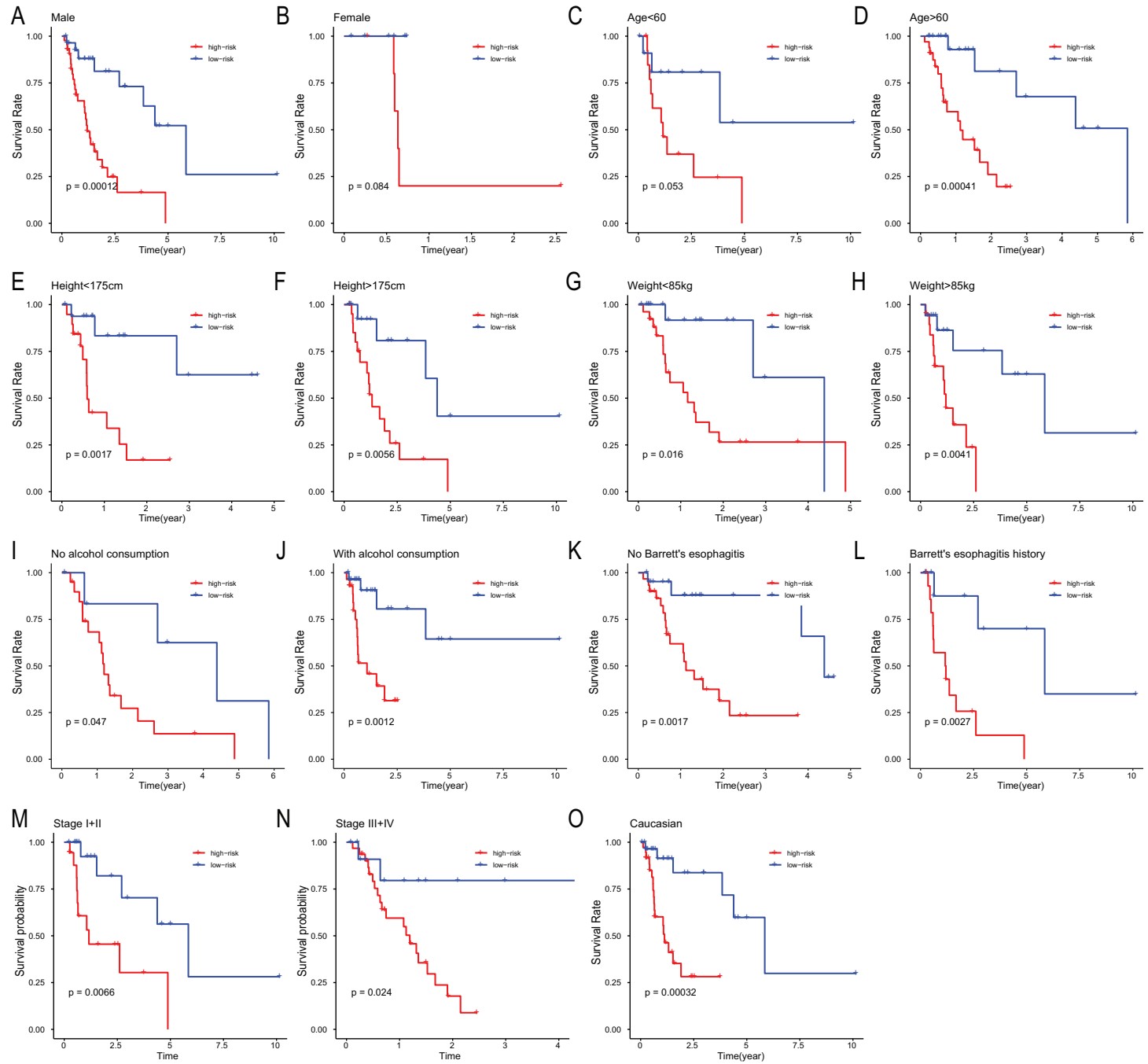

**Figure 4 Stratified analysis of overall survival in the entire set.** Kaplan–Meier analysis for OS in subgroups stratified by gender (A, B), age (C, D), height (E, F), weight (G, H), alcohol consumption (I, J), Barrett's esophagitis (K, L), TNM stage (M, N), caucasian (O).

significantly, which can be attributed to the lack of reliable biomarkers and genetic signatures for proper individualized treatment. Therefore, it is urgent to build the molecular signature of EAC to improve the survival rate and tailor effective personalized treatment. A large number of studies reported that miRNAs can play a key role in the diagnosis of tumors, the prediction of chemotherapy efficacy, and the prediction of cancer

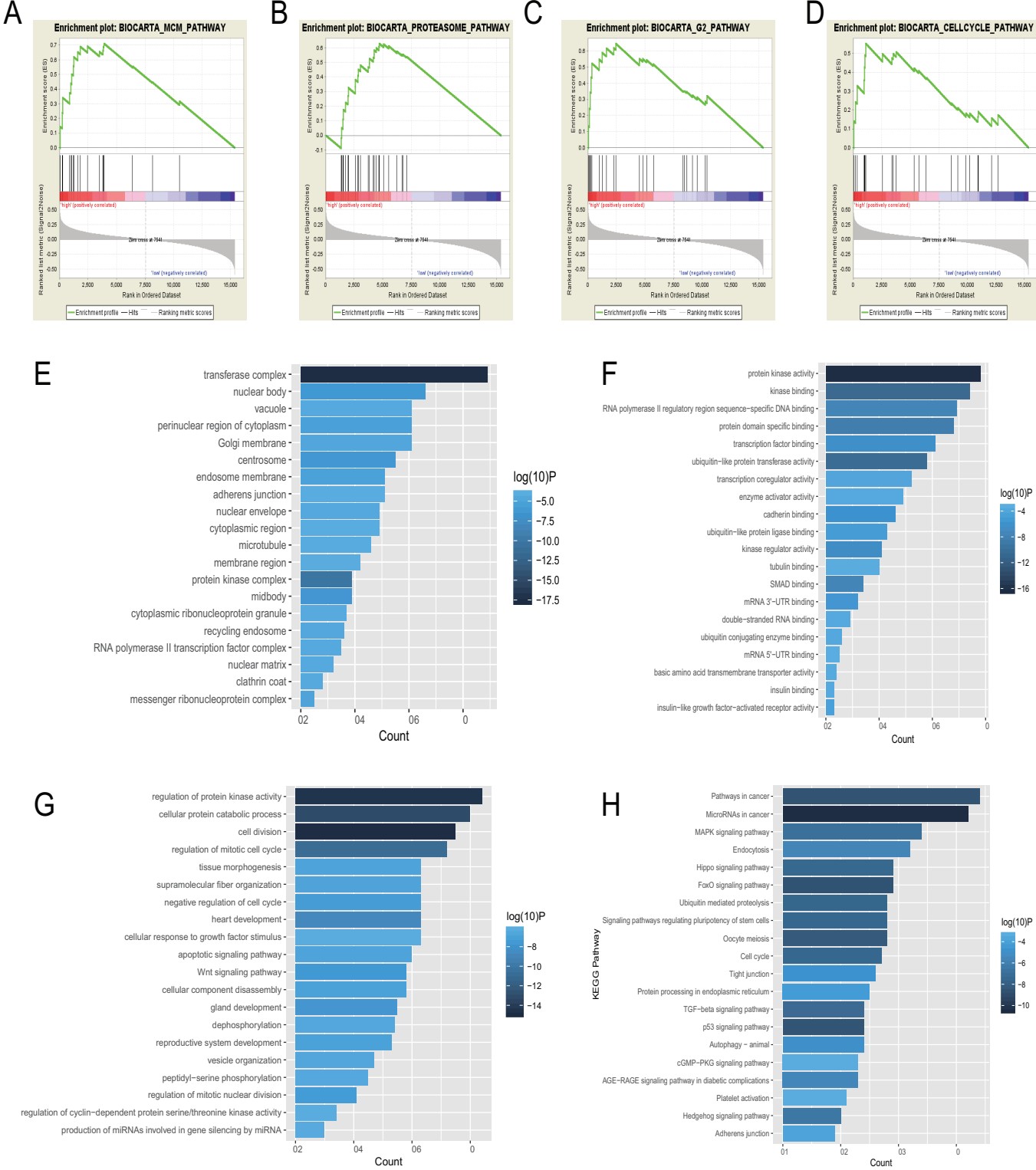

**Figure 5 Gene enrichment analysis, GO, and KEGG pathways of mRNA associated with the 6-miRNA signature.** (A–D) Gene sets enriched in high risk patient group compared to low risk group. The cellular component (E), molecular function (F) and biological process (G) of GO of the target genes. (H) The bar chart of significantly KEGG pathways of the target genes.
risk (*Mari et al., 2018*). The miRNAs have been reported to predict Barrett's disease development to EAC, diagnosis, prognosis, and treatment effect in EAC (*Maru et al., 2009*; *Nguyen et al., 2010*; *Wang et al., 2016*; *Zhang et al., 2013*). Data mining of TCGA is an effective way to identify genetic alterations related to clinical outcomes and screen novel therapeutic targets. In the last decade, miRNAs have attracted increasing attention in cancer research. However, very few studies have assessed the prognostic value of miRNA signature for patients with EAC on the basis of TCGA data. In this study we used univariate Cox regression analyses to identify 64 miRNAs, among which six miRNAs were selected to construct the risk score system for EAC prognosis through LASSO.

Through our analysis, we suggested that hsa-let-7b and hsa-mir-23a may enhance the survival rate of EAC patients, while hsa-mir-3074, hsa-mir-424, hsa-mir-425, and hsa-mir-505 may reduce the survival rate of EAC patients. Previous research has identified hsa-let-7b as a prognostic marker in NSCLC (*Hosseini et al., 2018*). Importantly, hsa-let-7b has been reported to inhibit cell proliferation, migration, and invasion in various malignant tumor by targeting different proteins (*He et al., 2018*; *Xu et al., 2014*; *Yu et al., 2015*). It was reported that hsa-mir-23a played various roles in the initiation, progression, diagnosis, prognosis, and treatment of tumors (*Wang et al., 2018*). Meanwhile, hsa-mir-23a was associated with differentiation and carcinogenic process of esophageal squamous cell cancer (*Zhu et al., 2013*).

Few studies have been published on the function of hsa-mir-3074 in carcinogenesis; therefore, it deserves further investigation. Hsa-mir-424 was recognized to play a dual role in various cancers. In colorectal cancer, hsa-mir-424 was identified as a tumor suppressor as it inhibited cancer cell growth and enhanced apoptosis (*Fang et al., 2018*). In addition, hsa-mir-424 was upregulated and correlated with poor survival in esophageal squamous cell carcinoma; it can promote cell proliferation by multilayered regulation of cell cycle (*Wen et al., 2018*).

The impact of hsa-mir-425 and hsa-mir-505 on other cancers seems to differ from its effect on EAC based on our bioinformatics analysis. A recent study indicated that hsa-mir-425 inhibited lung adenocarcinoma cellular proliferation and promoted cell apoptosis (*Liu et al., 2018*). Hsa-mir-425 can also inhibit cell proliferation of renal cell carcinoma by targeting E2F6 (*Cai et al., 2018*). Meanwhile, several articles have reported that hsa-mir-505 suppresses cell proliferation and invasion by targeting certain mRNAs in endometrial carcinoma and gastric cancer (*Chen et al., 2016*; *Tian et al., 2018*). However, overexpression of hsa-mir-425 and hsa-mir-505 was a poor prognostic factor in this study, and they may play a role as oncogenes of EAC.

Functional annotations in high-risk patients with EAC revealed that minichromosome maintenance (MCM) pathway, G2 pathway, and cell cycle pathway were enriched significantly. There are 10 proteins in the family of MCM complex, named MCM1-10 (*Nowinska & Dziegiel, 2010*). It has been reported that MCM2-7 play an important role as the eukaryotic replicative helicase through unwinding DNA and traveling with the fork (*Bochman & Schwacha, 2008*; *Labib, Tercero & Diffley, 2000*), along with the cyclin-dependent kinases as master regulators of the cell cycle and the initiator proteins of DNA replication, such as the origin recognition complex, Cdc6/18 (*Chen, de Vries & Bell, 2007*;

*Diffley et al., 1994*). There is evidence that high expression of MCM4 and MCM7 were associated with lymph node metastasis and shorter survival in EAC (*Choy et al., 2016*). Based on the result of GSEA, molecular function of GO, and KEGG, the 6-miRNA signature may be involved in regulation of cell cycle and DNA replication.

This study has certain limitations. First, the initial screening univariate Cox regression analyses included only 272 miRNAs after elimination of very low expressed miRNAs, whereas more than 4,000 human miRNAs have been discovered at present (*Chou et al., 2018*). Although the 6-miRNA signature can predict prognosis of EAC well, other miRNAs which have good predictive power for prognosis may have been missed. Second, due to the patient number limitation of TCGA, there are only 84 EAC patients, and fewer number of patients were included in subgroup analyses. Third, there were no external validation cohorts in this study which can convincingly validate the miRNA signature. Therefore, further studies will be needed to validate these findings using larger numbers of patients, and to explore potential molecular functions of the six separate miRNAs in EAC.

## CONCLUSIONS

In summary, we constructed a novel 6-miRNA-expression-based risk model based on TCGA dataset which displayed the potential to be an independent prognostic factor for patients with EAC. In addition, the miRNA signature can help improve our understanding of clinical decision-making as potential biomarkers and targets for patients with EAC.

## ACKNOWLEDGEMENTS

This study is based on data from the Cancer Genome Atlas (TCGA) database.

### Funding
The authors received no funding for this work.

### Competing Interests
The authors declare that they have no competing interests.

### Author Contributions
- Tian Lan conceived and designed the experiments, performed the experiments, analyzed the data, prepared figures and/or tables, authored or reviewed drafts of the paper, approved the final draft.
- Yunyan Lu prepared figures and/or tables, approved the final draft.
- Zunqiang Xiao contributed reagents/materials/analysis tools, approved the final draft.
- Haibin Xu analyzed the data, approved the final draft.
- Junling He prepared figures and/or tables, approved the final draft.
- Zujian Hu performed the experiments, contributed reagents/materials/analysis tools, approved the final draft.
- Weimin Mao conceived and designed the experiments, approved the final draft.

## Human Ethics

The following information was supplied relating to ethical approvals (i.e., approving body and any reference numbers):

All data were downloaded from the Cancer Genome Atlas (TCGA), therefore ethical approval was not needed.

## Data Availability

RNA-seq data and associated clinical information were downloaded from the TCGA data portal and are available as Supplemental Files.

## Supplemental Information

Supplemental information for this article can be found online at http://dx.doi.org/10.7717/peerj.7353#supplemental-information.

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
