# Peer review of "A six-microRNA signature can better predict overall survival of patients with esophagus adenocarcinoma"

_PeerJ, doi:10.7717/peerj.7353_

## Round 0.1 · original submission · Major Revisions

Dear Dr. Mao,

Please address all issues raised by the expert reviewers.

I agree that there are several typos, grammar errors throughout the manuscripts that need to be fixed.

Moreover, it should be clarified in the manuscript that the EAC cases whose data were analyzed are diagnostic.

Importantly, the miRNA expression signature identified in this study is not related to therapy reponse; hence, "prognostic" should be used instead of "predictive" when you refer to the biomarker identified in this study.

All of the changes and comments indicated in the attached annotated manuscript should also be addressed.

Best Regards,

[]

Reviewer 1 ·

Basic reporting

In this study, the authors aim to construct a miRNA-based signature for predicting the prognosis of esophagus adenocarcinoma (EAC) based on the data extracted from The Cancer Genome Atlas (TCGA). The authors downloaded RNA-seq data of 84 EAC cases from the TCGA database. The manuscript might be interesting for the clinicians who are mainly interested in the prognosis of EAC. However, the paper needs some additional works, in particular the statistical methods parts, and some major changes are recommended before considering the paper for publication. The manuscript also needs extensive English editing since there are several typos and grammatical errors.

Experimental design

In addition to this, the similarity index in the Turnitin Originality Report was 38%, and this value exceeded the acceptable level range between 15%-20%. Authors must be careful about citing papers appropriately in their study.

Validity of the findings

My concerns are as follows:

In abstract,

• In Line 57: Change “The Area under curve (AUC) for ROC” to “The area under the curve (AUC) for ROC” or “The area under the ROC curve”.
• In Line 58: Use a comma before the word “respectively”.
• In Line 63-64: It may be good to give a conclusion as the last sentences of the abstract.

In introduction,

• In Line 57: Change “More often, There …” to “More often, there …”.

In materials and methods,

• In Line 111: Provide the reference and accession numbers for the datasets.
• In Line 113: Correct “, Some miRNAs” to “, some miRNAs”.
• In Line 120: Change “metastasis status, TNM stage” to “metastasis status, and TNM stage”.
• In Line 121: Although the authors revealed that they used 84 EAC cases, they mentioned in the discussion section (in line 294) there were 87 EACs. They also used totally 84 patients in Table 1. Please check your results and be consistent in the number of total cases.
• In Line 126: How the authors randomly divided the data into two groups as train and test sets. Which computer program did they use? Why they preferred to split the data 50% rather than 70%-30% or 60%-40%?
• In Line 131: Correct “algorism” to “algorithm”.
• In Line 135-137: Did the authors use statistically significant genes to calculate risk score in the formula?
• In Line 136: Show the formula with a formula number (1) appropriately.
• In Line 139-153-180-195: The authors mentioned the same approach four times in different parts. Please summarize what you did only one time in the methods section. In addition, why the authors prefer to use the median value as a cut-off? Why did they not try to find any cut-off value with ROC to divide patients into the high and low risk groups?



In Results,

• In Table 1: Use tab for categories to make the table more readable. In addition to this, recalculate the percentages of each variable, because total percentage exceeds 100% (height 104.8%, tumor size 100.1%, lymph node status 100.1%, metastases 100.1%, stage 100.1%). Correct “femal” and “metastsis”. Delete “(%)” after the words “metastasis” and “stage”.
• In Table 2: Is risk score categorical variable? If yes, please give the categories in parentheses (high/low) as in other categorical variables. Plus, please use * for p<0.05 and write the phrase “*p<0.05, statistically significant” in the footnote of the table.
• In Figure 3-4: Use 3 digit for decimals of the p-value in Fig 3(D) (such as p-value<0.001). Which test results were shown with p-values in these figures? Please give the name of the test method in the materials and methods section.
• In Line 207: As I understood, the authors did not use correlation analysis, however they mentioned: “there was an independent correlation with OS after adjustment for other clinical pathological factors”. If you use correlation analysis, please add correlation analysis to the materials and methods section, else update the sentences.
• In Line 208-209: In Figure 4, “race” and “TNM stage” was not available in the table. However, there is “caucasion” in Figure 4G. Please update the sentences in these lines or add tables for race and TNM stage.
• In Line 213: “Moreover, this signature seemed more applicable to male Caucasian patients over 60.” How the authors get this result, please mention in the manuscript which figure shows this result.

In Discussion,

• In Line 240: Correct “paly” to “play”.
• In Line 261: Change “about The” to “about the”.
• In Line 293: Correct “Second, Due” to “Second, due”.

Annotated reviews are not available for download in order to protect the identity of reviewers who chose to remain anonymous.

Reviewer 2 ·

Basic reporting

The language requires considerable improvement. There are numerous places where tense is used incorrectly and syntax errors exist. 2. Many Hsa-miR names are written as Has-miR. Please correct those.

Experimental design

Please describe more clearly how the 6 miRNAs were selected from the 64 miRNAs that had predictive value.

The primary deficiency of the manuscript is the lack of a validation set. While the training and the test sets are fine, the lack of validation makes the study preliminary. It is suggested that the 6 miRNA signature is validated with an independent total RNA or small RNA/circulating miRNA sequencing dataset with clinical parameters for esophageal cancer.

Validity of the findings

The data needs to be validated with a validation set to be meaningful. Please see comments under experimental design.

---

## Round 0.2 · Minor Revisions

Dear Dr. Mao,

Your manuscript needs to have a minor revision.

The comments and modifications indicated in the attached annotated manuscript should be addressed.

In addition, the x-axes of Figures 2A, 2C, 3A-D should be labelled.

Best Regards,

Reviewer 1 ·

Basic reporting

no comment

Experimental design

no comment

Validity of the findings

no comment

Reviewer 2 ·

Basic reporting

No comment

Experimental design

No comment

Validity of the findings

No comment

Additional comments

The authors have appropriately addressed my concerns.

---

## Round 0.3 · accepted · Accept

Dear Dr. Mao,

Your manuscript may be acceptable as long as all the issues indicated in the annotated manuscript are addressed.

Best Regards,